# Factors Associated with Single-Use and Co-Use of Tobacco and Alcohol: A Multinomial Modeling Approach

**DOI:** 10.3390/ijerph16193506

**Published:** 2019-09-20

**Authors:** Jin-Won Noh, Kyoung-Beom Kim, Jooyoung Cheon, Yejin Lee, Ki-Bong Yoo

**Affiliations:** 1Department of Healthcare Management, Eulji University, Seongnam 13135, Korea; jinwon.noh@gmail.com (J.-W.N.); yiye1110@gmail.com (Y.L.); 2Global Health Unit, Department of Health Sciences, University Medical Centre Groningen, University of Groningen, Groningen 9713 GZ, The Netherlands; 3Graduate School of Public Health, Korea University, Seoul 02841, Korea; aefile01287@gmail.com; 4College of Nursing, Sungshin Women’s University, 55 Dobong-ro 76ga-gil, Gangbuk-gu, Seoul 01133, Korea; jcheon@sungshin.ac.kr; 5Department of Health Administration, Department of Information & Statistics, Yonsei University, Wonju 26493, Korea

**Keywords:** tobacco smoking, alcohol consumption, co-user, multinomial modeling

## Abstract

Tobacco smoking and alcohol consumption are among the most important public health concerns not only in South Korea but also globally. This study identified the factors associated with single-use and co-use of tobacco and alcohol in Korean adults and provided more accurate estimates using a multinomial modeling approach. This study used the Korea Community Health Survey Data 2017, of which 205,336 respondents were selected as the sample for a multinomial logistic regression analysis. For the group that identified as only drinking monthly compared to the reference group, we found that the direction of the following factors was opposite to that of the results of the only currently smoking group: Age, marital status, educational level, monthly household income, occupation, obesity, self-rated health, hypertension, and hyperlipidemia. For the currently smoking and drinking monthly group relative to the reference group, the overall direction was a mix of the results of only currently smoking and only drinking monthly. These findings support the development of policies that consider the risk of smoking tobacco and consuming alcohol simultaneously.

## 1. Introduction

Tobacco smoking and alcohol consumption are among the most important public health concerns not only in South Korea but also globally; they are among the top five major risk factors of death worldwide [1,2]. The harm of tobacco smoking and alcohol consumption, especially as a risk factor of several diseases, has been demonstrated in many studies [2,3,4,5]. Several studies have suggested that tobacco smoking and alcohol consumption are often seen together and share a number of risk factors [6,7]. Tobacco smoking is also a powerful predictor of alcohol consumption in adults [8,9]. However, regarding the additive detrimental effects of alcohol-tobacco co-use, the literature offers mixed results. Numerous studies have supported its synergetic effects [10], whereas one study pointed out the insufficient evidence of synergetic effects [11].

Some researchers have suggested that the level of tobacco- or alcohol-related ill health is higher among people in deprived communities than in non-deprived communities, although both groups smoke and consume the same amount of tobacco and alcohol [12,13,14,15,16]. For example, smokers or drinkers living in low-income communities are more likely to combine health damaging behaviors, such smoking and drinking, compared with people in more affluent communities [17]. These combinations not only add to the dangers of alcohol consumption but also multiply the risks of ill health. Consequently, it could lead to worsening socioeconomic health disparities. However, only a few studies have examined the relation between co-use of tobacco and alcohol and socioeconomic status (SES).

Previous studies have analyzed the effect of tobacco use, alcohol use, and their co-use using a separate binomial approach (e.g., to fit three binomial logistic regression models with only tobacco smokers, only alcohol drinkers, and co-users of tobacco and alcohol) [18]. However, this approach is suboptimal for several reasons, such as the occurrence of multiple testing problems and loss of information. These limitations could be overcome through a multinomial modeling framework [19]. Therefore, this study analyzed the factors associated in single-use and co-use of tobacco and alcohol using a multinomial modeling approach.

Tobacco smoking and alcohol consumption are among the representative and modifiable lifestyle risk factors. However, it is still unclear not only what role that demographic, socioeconomic, and health related factors have on it but also the combined associations of tobacco smoking and alcohol consumption. Therefore, greater understanding of single-use and co-use of tobacco and alcohol may improve population health and decrease societal burden. Moreover, many major findings about the current issue have been observed in developed Western countries [20,21,22], but little is known about Asian populations [23,24]. This study investigated the factors associated in single-use and co-use of tobacco and alcohol in Korean adults using data from a nationwide representative survey. This study also provided more accurate estimates using a multinomial modeling approach. Moreover, this study extended the existing knowledge to groups of only currently smoking, only drinking monthly, and currently smoking and drinking monthly.

## 2. Materials and Methods

### 2.1. Study Population and Data

This study used the Korea Community Health Survey Data 2017. This survey included SES, health status, medical utilization, and health behaviors. This survey was conducted in 253 community health centers in Korea [25]. The purpose of this survey was to produce regional representative statistics in Korea; its target population was general Korean adults aged 19 years or above. The primary sampling unit was Tong, Ban, Lee, which are the smallest administrative district units in Korea. The secondary sampling unit was households. Tong, Ban, Lee were selected through probability proportionate sampling. Households were selected using systematic sampling. When a household had been selected as a sample, trained interviewers visited the household to conduct one-on-one interviews. In the 2017 dataset, the number of respondents was 228,381. After we excluded missing variables (*n* = 23,045), our study population was reduced to 205,336 respondents. 

### 2.2. Variables

The dependent variable was the combined variable of currently smoking and monthly drinking. If respondents answered daily or sometimes smoking and smoking more than five packs in their entire life, then they were categorized as currently smoking. Monthly drinking was defined as whether respondents had drunk more than once a month during the last year or not. Based on the definitions of currently smoking and monthly drinking, the dependent variables were categorized into the following: 0 = Not currently smoking/not drinking monthly, 1 = currently smoking/not drinking monthly, 2 = not currently smoking/drinking monthly, 3 = currently smoking/drinking monthly.

Independent variables were age, sex, marital status, educational level, household income, occupation, region, depressive symptoms, obesity, self-rated health, hypertension, diabetes, and hyperlipidemia. Marital status was categorized into married, divorce/separated/bereaved, and single. Education was categorized under elementary school, middle school, high school, and university and higher. Household income was categorized into USD 999 and below, USD 1000–USD 2999, USD 3000–USD 5000, USD 5000 and above. Approximately KRW 1000 were equivalent to USD 1. Occupation was categorized into managers, office workers, sales and service workers, skilled agricultural/forestry/fishery workers, elementary workers, and others. Others included students, housewives, and unemployed people. Region was categorized into city and rural. Depressive symptoms were derived from the Patient Health Questionnaire depression module (PHQ-9). Obesity was categorized into underweight (body mass index (BMI) < 18.5), normal (18.5 ≤ BMI < 25.0), overweight (25.0 ≤ BMI < 27.0), and obese (BMI ≥ 27.0). Self-rated health was categorized into very bad, bad, normal, good, and very good. Hypertension, diabetes, and hyperlipidemia were derived from the question “Do you currently receive treatment for hypertension/diabetes/hyperlipidemia?” 

### 2.3. Statistical Analysis

As the dependent variable consisted of four groups, a multinomial logistic regression analysis was selected to identify the associations between independent variables and smoking and drinking status. All independent variables were included in the model. All analyses were conducted in SAS version 9.4 (SAS Institute Inc., Cary, NC, USA). The significance level used was 0.05.

### 2.4. Research Ethics

All participants gave their informed consent for inclusion before they participated in the study. The study was conducted in accordance with the Declaration of Helsinki. The protocol was approved by the Ethics Committee of Eulji Institutional Review Board (EUIRB2018-68).

## 3. Results

Participants’ general characteristics are shown in Table 1. Among the 205,336 respondents, 90,355 (44.0%) answered not currently smoking and not drinking monthly, 8608 (4.2%) were only currently smoking, 79,422 (38.7%) were only drinking monthly, and 26,951 (13.1%) were currently smoking and drinking monthly.

Table 2 presents the results of the multinomial logistic regression. Not currently smoking and not drinking monthly group served as the reference group for the dependent variable. Results showed that most independent variables were significantly associated with all the dependent groups compared with the reference group. Moreover, the following factors increased the odds of being a smoker relative to the reference group: Being younger, being male, having a low education level, having a low monthly household income, being not married, having depressive symptoms, having low BMI, having self-rated one’s health as bad, being hyperlipidemic, and living in a city. For only drinking monthly relative to the reference group, the direction of the following factors was opposite that of those for only currently smoking: Age (30–39 and 40–49), marital status (single), educational level (middle school, high school, and university and higher), monthly household income (USD 1000–USD 2999, USD 3000–USD 5000, and USD 5000 and above), occupation (others), obesity (normal, overweight, and obese), self-rated health (normal and bad), hypertension, and hyperlipidemia. For currently smoking and drinking monthly group relative to the reference group, the overall direction was a mix of the results of only currently smoking and only drinking monthly. The only difference was that the odds ratios of only currently smoking were lower than those of drinking monthly in aged 60–69, aged 70 and above, being female, other occupation, living in rural areas, having diabetes.

## 4. Discussion

Recent research findings have directed public health attention toward bad lifestyle habits, especially smoking and drinking. Previous studies have described that these two lifestyle habits are associated with each other [6,7] and emphasized that the co-use of tobacco and alcohol is a greater risk than the sum of their independent effects [6,7,26]. However, only a few studies have examined the factors associated with co-use of tobacco and alcohol compared with the use of tobacco only and the use of alcohol only. This study extended the existing knowledge to groups of only currently smoking, only drinking monthly, and currently smoking and drinking monthly.

This study found that adults in their thirties to fifties had a high risk of currently smoking and currently smoking/drinking monthly, which is largely consistent with previous studies [6,18,27,28]. A previous study noted that current middle-aged smokers may have multiple tobacco-related health problems that may possibly result in earlier mortality [18], indicating the need for preventive interventions that promote smoking cessation for this age group. Most smokers have been found to start smoking during their adolescence, but young adulthood is considered a critical period in the progression towards an established smoking behavior and nicotine dependence, especially among males [28,29,30]. Many adults view themselves as “social smokers” by which they consider their smoking behaviors as a social interaction in young adulthood [31]. 

The notable finding of this study is that being aged 50 was a turning point of the direction of the relation in the only currently smoking and currently smoking/drinking monthly groups. Meaning, middle-aged adults were concerned about their health and prepare for retirement and later life. The literature has noted that most adults have regular check-ups in their 40s, begin to recognize age-related changes, or are diagnosed with chronic diseases and thus need medications [32,33]. Previous studies have indicated that smoking cessation and limiting alcohol consumption are always beneficial in regard to increasing life expectancy and decreasing mortality and health costs [6,34]. Therefore, educational interventions regarding good lifestyle habits and smoking cessation should be provided for adults in their 40s.

This study showed that adults in their 20s had a high risk of being a monthly drinker. The literature identified developing interpersonal relationships, relieving negative emotions due to stress and personal problems, and attending social activities, such as festivals, celebrations, and meetings as the major reasons for drinking [35,36,37]. Compared with other countries, the drinking culture among college students, especially among men, in Korea is relatively permissive and more associated with social drinking with friends and colleagues; this culture tends to encourage excessive alcohol consumption [27,35,36,37,38,39]. The Korean drinking culture of encouraging excessive alcohol consumption and coercive offers should be changed. Effective interpersonal communication interventions should be developed to promote healthier activities with friends and colleagues, as well as prevent harmful social drinking practices [35,36,37,38]. 

Previous studies have reported that living alone is more likely associated with multiple lifestyle risk factors, such as smoking and heavy alcohol drinking [27,40,41,42,43], even when gender difference is considered [44]. In this study, divorced/separated/bereaved adults were more likely to be currently smoking and currently smoking/drinking monthly compared with married adults; this finding supports previous findings [41,42,43]. Harmful lifestyle habits, such as smoking or heavy alcohol drinking, may be more noticeable by spouses and may be changed through the support and encouragement of spouses, especially wives [41,42,44]. A spouse is the most influential person to a smoker or drinker as he/she is the person with whom the latter is in a romantic relationship; therefore, a couple-based intervention that helps adults quit smoking and prevent heavy alcohol drinking, may be more helpful in changing negative health behaviors, especially among men [43].

Interestingly, in this study, single adults were less likely to be drinking monthly compared with married adults. Watt et al. (2014) reported that heavy drinking is more common among divorced/separated/bereaved men and single women in the United States (U.S.) [42]. By contrast, Li et al. (2017) found that living with a spouse or partner is related to a higher risk of alcohol consumption in Norway. The Korean drinking culture is more associated with social drinking with others compared with other countries; therefore, drinking alone at home may not be common among single adults [37,38,39]. The findings in this study highlighted the importance of living arrangements to smoking and drinking habits and provided evidence for the use of different approaches for men and women by marital status. 

The literature has offered mixed results on the interactions between smoking and drinking habits according to SES in Korea. A study reported that the prevalence of smoking and drinking is higher in lower SES groups [45], whereas other studies found that educational level is a significant factor that is associated with drinking [40] and income does not significantly affect alcohol drinking [40] and smoking [46]. Kang and colleagues found that educational and income levels are not associated with smoking habits [28].

In this study, low and high educational status were associated with currently smoking and drinking monthly, respectively. Among currently smoking and drinking monthly group, adults who graduated from middle school and high school were more likely to be currently smoking and drinking monthly, whereas those who graduated from a university were less likely to be a co-user of tobacco and alcohol, compared with those who did not graduate elementary school. The findings may be interpreted in terms of coping strategies to stress, as well as availability and accessibility of health service and social networking. Smoking may be among the coping strategies of adults in low SES to deal with difficult and stressful situations [46]. However, adults in high SES groups may have more chances to access the health care system and more knowledge about the harmful health effects of tobacco use, which lead to more smoking cessation [46]. They may also have more alternatives to replace smoking as a coping strategy in a difficult and stressful situation. In the aspect of drinking, a higher educational level is positively associated with drinking monthly, indicating that adults in high SES can afford alcohol, consume more alcohol in social activities, and often drink alcohol to relax at leisure time [37,39]. 

The association between occupation stress and smoking/alcohol use can be explained on several grounds. Previous studies have reported that individuals can self-medicate stress-induced physiological effects by smoking/drinking to achieve internal stability [47]. Muraven (2000) explained that job stress can reduce an individual’s self-control, making it difficult for current smokers/drinkers to quit or reduce smoking/drinking intensity and may induce former smokers/drinkers to relapse and start smoking/drinking again [48,49]. This study found consistent evidence on the association between occupation and smoking/alcohol use. Compared with other groups, all occupational groups showed a high level of smoking and drinking.

The relations of perceived health status to currently smoking, drinking monthly, and currently smoking and drinking monthly were interesting in this study. Perceived normal and bad health status were associated with currently smoking, whereas perceived bad health status was related to lower odds of drinking monthly. Adults who perceived their health as being normal were more likely to be currently smoking and drinking monthly. By contrast, those who rated their health poorly were less likely to co-use tobacco and alcohol. However, the present study’s findings could not clarify the direction of causality in the relation. Perceived poor health status may influence either the decisions to stop drinking or the use of tobacco as a means to alleviate stress and pain [6,27,37]. Therefore, additional studies are needed to examine the causal relation between two lifestyle habits and perceived health status. 

Previous studies have found inconsistent results on regional differences between smoking and drinking habits in other counties, such as Finland, the U.S., and China [50]. In this study, people living in cities were more likely to be currently smoking and drinking monthly compared with those in rural areas. Various factors, such as social norms, family status and relationships, education, SES, and religious beliefs, have been attributed to these differences [50].

This study has several limitations. First, this study used a cross-sectional design, which limits any consideration of causal relations, such as that between health-related factors and two lifestyles habits. Second, self-reported data on smoking and drinking may not provide accurate information because of recall and social desirability biases. Third, the amounts and frequency of smoking and drinking were not examined in detail; this may influence the interpretation of study findings. For example, light to moderate alcohol consumption may reduce the risk of some diseases [51]. Finally, psychological factors, such as stress, self-efficacy, and social support, could not be adjusted.

In spite of these limitations, this study holds several strengths. This study used a nationally representative sample and included a large sample size. Analyses were performed separately to examine the different factors associated with each of the four groups (not currently smoking/not drinking monthly, only currently smoking, only drinking monthly, currently smoking/drinking monthly). In this way, this study provided evidence for developing tailored interventions and strategies for specific vulnerable groups.

## 5. Conclusions

Using a nationwide representative survey, this study showed the importance of living arrangements to tobacco smoking and alcohol consumption and provided evidence for using different approaches depending on demographic, socioeconomic, and health-related factors. These findings support the importance of policies considering the risk of smoking tobacco and consuming alcohol simultaneously. Findings suggest that policy makers and health care professionals must have better knowledge of Korean cultural aspects in relation to age, marital status, and SES, such as social smoking and drinking for social networking.

## Figures and Tables

**Table 1 ijerph-16-03506-t001:** Participants’ general characteristics.

Variable	Category	Not Currently Smoking/Not Drinking Monthly	Only Currently Smoking	Only Drinking Monthly	Currently Smoking and Drinking Monthly
Age (years)	19–29	6360 (27.3)	717 (3.1)	12,629 (54.1)	3630 (15.6)
	30–39	7613 (27.6)	1046 (3.8)	13,549 (49.1)	5398 (19.6)
	40–49	10,364 (29.4)	1530 (4.3)	16,517 (46.8)	6870 (19.5)
	50–59	15,751 (39.9)	1836 (4.7)	15,714 (39.8)	6145 (15.6)
	60–69	19,025 (53.4)	1666 (4.7)	11,601 (32.6)	3308 (9.3)
	70 and above	31,242 (70.9)	1813 (4.1)	9412 (21.4)	1600 (3.6)
Sex	Male	21,918 (24)	7223 (7.9)	37,054 (40.6)	24,996 (27.4)
	Female	68,437 (60)	1385 (1.2)	42,368 (37.1)	1955 (1.7)
Marital status	Married	59,318 (43.2)	5339 (3.9)	55,491 (40.4)	17,288 (12.6)
	Divorce/separated/bereaved	22,252 (63.8)	1668 (4.8)	8037 (23.1)	2896 (8.3)
	Single	8785 (26.6)	1601 (4.8)	15,894 (48.1)	6767 (20.5)
Education level	Under elementary school	35,889 (69.1)	2213 (4.3)	11,281 (21.7)	2548 (4.9)
	Middle school	11,083 (49.8)	1191 (5.3)	7408 (33.3)	2589 (11.6)
	High school	23,358 (34.3)	3237 (4.8)	29,255 (43)	12,200 (17.9)
	University and higher	20,025 (31.7)	1967 (3.1)	31,478 (49.9)	9614 (15.2)
Monthly household income (USD)	999 and below	27,191 (65.8)	2154 (5.2)	9141 (22.1)	2842 (6.9)
	1000–2999	30,781 (45.3)	3300 (4.9)	24,214 (35.6)	9627 (14.2)
	3000–5000	19,760 (34.8)	2007 (3.5)	25,923 (45.7)	9014 (15.9)
	5000 and above	12,623 (32.1)	1147 (2.9)	20,144 (51.2)	5468 (13.9)
Occupation	Managers	7027 (31.2)	682 (3)	11,373 (50.5)	3429 (15.2)
	Office workers	4296 (23.9)	479 (2.7)	10,216 (56.7)	3017 (16.8)
	Sales and service workers	9304 (35.2)	1026 (3.9)	12,039 (45.6)	4039 (15.3)
	Skilled agricultural/forestry/fishery workers	12,544 (51.2)	1280 (5.2)	7950 (32.4)	2728 (11.1)
	Elementary workers	11,532 (30.7)	2381 (6.3)	14,294 (38)	9378 (25)
	Others	45,652 (59.8)	2760 (3.6)	23,550 (30.9)	4360 (5.7)
Depressive symptoms	No	86,607 (43.6)	8121 (4.1)	77,725 (39.1)	26,123 (13.2)
	Yes	3748 (55.4)	487 (7.2)	1697 (25.1)	828 (12.2)
Obesity	Underweight	10,918 (62.2)	738 (4.2)	4894 (27.9)	992 (5.7)
	Normal	58,393 (43.2)	5455 (4)	53,992 (39.9)	17,428 (12.9)
	Overweight	11,582 (40.1)	1266 (4.4)	11,493 (39.7)	4580 (15.8)
	Obese	9462 (40.1)	1149 (4.9)	9043 (38.3)	3951 (16.7)
Self-rated health	Good	26,402 (34.4)	2918 (3.8)	35,896 (46.7)	11,608 (15.1)
	Normal	35,540 (41.6)	3368 (3.9)	34,491 (40.3)	12,087 (14.1)
	Bad	28,413 (66)	2322 (5.4)	9035 (21)	3256 (7.6)
Hypertension	No	59,893 (39.2)	6747 (4.4)	63,790 (41.7)	22,543 (14.7)
	Yes	30,462 (58.2)	1861 (3.6)	15,632 (29.9)	4408 (8.4)
Diabetes	No	75,359 (41.9)	7590 (4.2)	71,814 (40)	24,987 (13.9)
	Yes	14,996 (58.6)	1018 (4)	7608 (29.7)	1964 (7.7)
Hyperlipidemia	No	78,120 (42.4)	7354 (4)	74,011 (40.1)	24,942 (13.5)
	Yes	12,235 (58.5)	1254 (6)	5411 (25.9)	2009 (9.6)
Region	City	56,068 (40.2)	5564 (4)	58,128 (41.7)	19,698 (14.1)
	Rural	34,287 (52)	3044 (4.6)	21,294 (32.3)	7253 (11)

**Table 2 ijerph-16-03506-t002:** Results of the multinomial logistic regression (reference group = not currently smoker/not drinking monthly).

Variable	Category	Only Currently Smoker			Only Drinking Monthly			Currently Smoking and Drinking Monthly		
		OR	95% Low	95% High	OR	95% Low	95% High	OR	95% Low	95% High
Age (years)	19–29	1.000			1.000			1.000		
	30–39	1.855	1.647	2.090	0.801	0.760	0.843	1.405	1.305	1.513
	40–49	1.720	1.524	1.942	0.658	0.623	0.695	1.118	1.035	1.207
	50–59	1.011	0.891	1.148	0.430	0.406	0.455	0.541	0.499	0.587
	60–69	0.577	0.502	0.662	0.315	0.296	0.335	0.221	0.202	0.243
	70 and above	0.317	0.274	0.367	0.208	0.195	0.223	0.073	0.066	0.081
Sex	Male	1.000			1.000			1.000		
	Female	0.042	0.040	0.045	0.337	0.329	0.346	0.019	0.018	0.020
Marital status	Married	1.000			1.000			1.000		
	Divorce/separated/bereaved	2.346	2.193	2.509	1.033	0.999	1.068	2.082	1.968	2.203
	Single	1.325	1.213	1.448	0.822	0.786	0.860	1.038	0.977	1.102
Education level	Under elementary school	1.000			1.000			1.000		
	Middle school	0.919	0.847	0.998	1.131	1.086	1.179	1.105	1.030	1.184
	High school	0.855	0.791	0.924	1.254	1.207	1.303	1.202	1.128	1.281
	University and higher	0.538	0.490	0.591	1.157	1.108	1.209	0.779	0.725	0.836
Monthly household income (USD)	999 and below	1.000			1.000			1.000		
	1000–2999	0.925	0.863	0.991	1.091	1.054	1.130	0.987	0.931	1.047
	3000–5000	0.782	0.718	0.850	1.250	1.202	1.301	0.932	0.873	0.995
	5000 and above	0.755	0.685	0.831	1.407	1.347	1.468	0.899	0.837	0.965
Occupation	Managers	1.000			1.000			1.000		
	Office workers	1.125	0.991	1.277	1.417	1.351	1.487	1.356	1.264	1.454
	Sales and service workers	1.740	1.560	1.941	1.224	1.171	1.280	1.655	1.549	1.769
	Skilled agricultural/forestry/fishery workers	1.189	1.059	1.336	1.066	1.013	1.121	1.079	0.999	1.165
	Elementary workers	1.509	1.364	1.669	1.079	1.031	1.128	1.447	1.361	1.539
	Others	1.003	0.905	1.112	0.749	0.721	0.779	0.627	0.587	0.669
Depressive symptoms	No	1.000			1.000			1.000		
	Yes	1.899	1.700	2.121	0.957	0.897	1.022	1.850	1.676	2.043
Obesity	Underweight	1.000			1.000			1.000		
	Normal	0.707	0.647	0.772	1.217	1.169	1.267	0.939	0.867	1.018
	Overweight	0.625	0.562	0.694	1.192	1.135	1.251	0.799	0.731	0.873
	Obese	0.679	0.609	0.757	1.107	1.052	1.165	0.753	0.688	0.826
Self-rated health	Good	1.000			1.000			1.000		
	Normal	1.148	1.085	1.213	0.961	0.939	0.985	1.247	1.202	1.293
	Bad	1.243	1.158	1.335	0.594	0.574	0.615	0.930	0.880	0.984
Hypertension	No	1.000			1.000			1.000		
	Yes	0.645	0.605	0.688	1.141	1.108	1.175	0.952	0.907	0.999
Diabetes	No	1.000			1.000			1.000		
	Yes	1.031	0.953	1.115	0.963	0.929	0.997	0.829	0.778	0.882
Hyperlipidemia	No	1.000			1.000			1.000		
	Yes	1.238	1.150	1.332	0.836	0.804	0.870	0.901	0.847	0.959
Region	City	1.000			1.000			1.000		
	Rural	0.899	0.852	0.949	0.849	0.828	0.871	0.817	0.785	0.849

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
