# Peer review of "Factors Associated with Single-Use and Co-Use of Tobacco and Alcohol: A Multinomial Modeling Approach"

_ijerph, 2019, doi:10.3390/ijerph16193506_

Round 1
Reviewer 1 Report
See attached

Author Response
This is a study that make a great contribution to better understanding of public health issues by identifying factors associated with co-user of tobacco and alcohol in an Asian country as well as in general. I strongly recommend the Journal this paper be published with respect to its strengths including use of a nationwide representative sample of general population and employing an advanced statistical analysis different from previous studies as long as I know.
However, it would be better paper if some revisions were made in terms of suggestions followed.
- For introduction, purpose of this paper would be more focused. As it is, purpose of this paper seems like to identify factors associated with co-user of tobacco and alcohol rather than to examine role of socioeconomic status (SES) playing in being simultaneous user of smoking and drinking alcohol. Some explanations need to be given about role of SES around line 58.
Thank you for your comment. We added explanations about role of SES as below; “However, it is still unclear not only role of demographic, socioeconomic, and health-related factors on but also combined associations of tobacco smoking and alcohol consumption. Therefore, greater understanding of the independent and combined associations of tobacco smoking and alcohol consumption may helpful to improve the population health and decrease the societal burden.”- Also, for discussion it should have been focused on differences across SES of factors identified by analysis if the purpose described is being in active. Of course, current conclusion derived from result would be modified by either focusing on results as analyzed (i.e., focusing on factors of being a co-user of tobacco and alcohol without incorporating role of SES) or making additional analyses to be able to compare the factors identified of being a co-user across SES.
As your comments, we explained SES’s role as important factors of being a co-user of tobacco and alcohol, in line 197-216. "Studies have found no consistent evidence of interactions between smoking and drinking habits according to socioeconomic status (SES) in Korea. A study reported that prevalences of smoking and drinking were higher in lower SES groups [45], while other studies found that educational level was a significant factor associated with drinking [40] and income was not a significant factor of alcohol drinking [40] and smoking [46]. In a study of Kang and colleagues, both educational level and income were not associated with smoking habits [28]. In this study, low educational status was associated with current smoking, whereas high educational status was associated with drinking monthly. Among current smoker and drinking monthly group, adults who graduate middle school and high school were more likely to be current smoker and drinking monthly, whereas adults who graduate university were less likely to be co-user of tobacco and alcohol comparing with those who graduate under elementary school. The findings may be interpreted in terms of coping strategies to stress, availability and accessibility of health service, and social networking. Adults in low SES, smoking may be one of coping strategies to deal with the difficulty and stressful situations [46]. However, adults in high SES may have more chance to access healthcare system and more knowledgeable about the harmful health effects of tobacco use, which led to more smoking cessation [46]. They may also have more alternatives to replace smoking as a coping strategy in a difficult and stressful situation. In the aspect of drinking, higher level of education was positively associated with drinking monthly, which may indicate that they are able to afford alcohol, and more consume alcohol in social activities, and often drink alcohol to relax at leisure time [37,39].”- Argument made in line 148-150 seems to be out of context.
As your comments, we removed the line 148-150 of the manuscript.- It would be good for reader to have information in more detail about how to collect data.
The primary sampling unit was Tong, Ban, Lee, which are the smallest administrative district units in Korea. The secondary sampling unit was households. Tong, Ban, Lee was selected through probability proportionate sampling. The households were selected by systematic sampling. When a household was selected as the sample, the trained interviewers visited the household to conduct one-on-one interviews.- Some explanations would be needed with respect to reasons of inclusion of independent variables.
Thank you for your comment. This paper included each independent variables based on the discussion of previous researches papers and opinions as researchers. This background is stated in the introduction to the manuscript. “Some researchers suggested that people in deprived communities had higher levels of tobacco-related or alcohol-related ill health than people in non-deprived communities, despite the same amounts of tobacco smoking or alcohol consumption [12-16]. For example, smokers or drinkers living in low-income communities were more likely to combine health damaging behaviors such smoking and drinking than people in more affluent communities [17]. These combinations do not just add to the dangers from alcohol consumption but multiply the risks of ill health. Consequently, it could lead to worsening socioeconomic health disparities. However, little study on the relationship between co-use of tobacco smoking and alcohol drinking and SES has been conducted. Previous studies have focused the effect of tobacco use, alcohol use, and their co-use were analyzed with the separate binomial approach (e.g., to fit three binomial logistic regression models with only tobacco smokers, only alcohol drinkers, and co-users of tobacco and alcohol) [18]. However, this approach is suboptimal for several reasons such as multiple testing problems and the loss of information. These limitations could be overcome through multinomial modeling framework [19]. Therefore, this study analyzed the independent and combined associations of tobacco smoking and alcohol consumption using multinomial modeling approach. Tobacco smoking and alcohol consumption were one of representative and modifiable lifestyle risk factors. However, it is still unclear not only role of demographic, socioeconomic, and health-related factors on but also combined associations of tobacco smoking and alcohol consumption. Therefore, greater understanding of the independent and combined associations of tobacco smoking and alcohol consumption may helpful to improve the population health and decrease the societal burden. Moreover, many major findings about current issue were done in developed Western countries [20-22], but very little is known in Asian populations [23,24].”- typos in line 87
As your comments, we revised the typo, as ‘Occupation‘ to ‘occupation’.
Reviewer 2 Report
Dear authors
I find this paper largely descriptive, with very basic methods. For example, the analysis only uses four groups (binary drinking/smoking).
The writing also needs to be improved. Please find a native English speaker or an English editing company for professional language editing.
However, there is nothing scientifically wrong with this paper.
Author Response
Dear authors
I find this paper largely descriptive, with very basic methods. For example, the analysis only uses four groups (binary drinking/smoking).
Thank you for your consideration. As your comment, this study used the basic method instead of sophisticated methods. It is simple, but powerful in our opinion, because there has been no such attempt in Korea, as well as other countries in our knowledge. The role of this paper provides information of the current situation to policy makers and professionals and the results can be used as base evidence of further studies.The writing also needs to be improved. Please find a native English speaker or an English editing company for professional language editing.
Thank you for your comment. We checked the English language and style again.However, there is nothing scientifically wrong with this paper.
